# Characteristics of Soil Mites Communities Structure under Vegetation Vertical Gradient in the Shibing World Natural Heritage Property, China

Yuanyuan Zhou †, Qiang Wei †, Niejia Xiao, Ju Huang, Tong Gong, Yifan Fei, Zheng Shi and Hu Chen *

School of Karst Science, Guizhou Normal University, Guiyang 550001, China; gy2020zhouyy@163.com (Y.Z.); 19010170413@gznu.edu.cn (Q.W.); 20010170516@gznu.edu.cn (N.X.); 19010170400@gznu.edu.cn (J.H.); 21010170528@gznu.edu.cn (T.G.); 21010170527@gznu.edu.cn (Y.F.); 21020171657@gznu.edu.cn (Z.S.)
* Correspondence: gy_chenhu@163.com
† These authors contributed equally to this work.

**Abstract:** In montane environments, as elevation increases, the combination of hydrothermal factors changing and vegetation types changing can cause changes to the soil mite community. To reveal the influence of different vertical vegetation types on the structure and diversity of soil mite communities in the Shibing Karst World Natural Heritage Property, in September 2021, specimen collection and identification of soil mites were carried out under the four typical vegetation zones of coniferous broad-leaved mixed forests (CBF), evergreen broad-leaved forests (EBF), deciduous broad-leaved forests (DBF), and river beach scrubs (RBS) in the Heritage Property. This occurred in order to analyze the community structure of soil mites. A total of 10,563 soil mites were captured in this region, belonging to 3 orders, 67 families, 137 genera; *Perscheloribates* and *Scheloribates* are the dominant groups in the area. The number of soil mite genera (CBF > EBF > DBF > RBS) and the number of individuals (RBS > DBF > CBF > EBF) differed between vegetation types. The dominant soil mite genera were not entirely consistent, with the highest values for each soil mite community diversity parameter being in the EBF habitat. The number of soil mite genera and individuals differed among vegetation types in different soil layers. It showed an apparent aggregation towards the surface layer, with complex diversity and richness indices changes. The highest community similarity indices were found between CBF and DBF, which were moderately similar. The cluster analysis results further showed that soil mite communities differed in different vegetation zones and among the same vegetation zones. The predatory gamasid mite structure is mainly *r*-selective. The ecological groups of oribatid mites are all O-type in the number of groups and P-type in the number of individuals. *Lasiobelba*, *Nanhermannia*, *Tectocepheus*, and *Mochlozetes*, among others, represent the group of nutrient functions that make up the soil mites in the study area. The study shows that the soil mite community of the Shibing Karst World Natural Heritage Property is rich in groups and shows gradient differences with the vegetation spectrum, and based on the unique subtropical canyon karst habitat of the Heritage Property, the community structure of soil mites will be in the process of adaptation and dynamic change, so long-term dynamic monitoring and in-depth study of the soil mites community structure of the Heritage Property are needed.

**Keywords:** soil mites; community structure; karst forests; Shibing Karst World Natural Heritage Property; UNESCO



## 1. Introduction

World Heritage is a cultural and natural heritage of outstanding and universal value, recognized by UNESCO and the World Heritage Committee as the common heritage of all humanity, and natural heritage is treasured more than cultural heritage because of its fragility and non-renewability [1,2]. The Shibing Karst World Natural Heritage Property (hereafter referred to as the Heritage Property), as one of the South China Karst II series

properties, is highly representative of the global distribution of dolomite karst landscapes in the tropical–subtropical region, and was inscribed on the World Heritage List at the 38th World Heritage General Assembly in June 2014 for meeting World Heritage Criterion (vii) and Standard (viii) [3]. The terrain of the Heritage Property is high in the north and low in the south, ranging from 593 m to 1246 m above sea level, with an average altitude of 912 m. The rugged terrain and the unique dolomite geomorphology have created different hydrothermal environments from the valley to the summit, forming a primitive subtropical evergreen broadleaf forest ecosystem in the lower and middle parts of the mountain, and a montane evergreen sclerophyll forest and coniferous broad-leaved mixed forest on the ridges and top of the hill. In terms of biodiversity, the vegetation has reached the top stage of natural succession, with a clear vertical vegetation spectrum and a complete community structure and ecological function, with a strong and relatively stable autochthonous nature, reflecting the unique biological and ecological, evolutionary characteristics of the Dolomite Karst Canyon [4,5]. Current studies on the biodiversity of karst forest ecosystems in Heritage property have mainly focused on aboveground surveys [5–7], and there is an apparent lack of research on the biodiversity of belowground ecosystems. This is especially true of the study of soil organisms under different vegetation types, with only Zhang Yan [8] reporting on the structural characteristics of soil mite communities in the area, which seriously limits the understanding of the diversity of karst forest subsurface ecosystems; therefore, comparative studies on soil mite community structure and diversity under different vertical gradients of forest vegetation in the Heritage Property are essential for understanding changes in above- and belowground ecological processes.

Soil mites are an essential component of the forest soil environment and play a crucial role in the decomposition of soil organic matter, promotion of soil fertility, participation in material cycling, and plant community succession [9–12]. In recent years, there has been increasing interest in the interaction of soil mites with aboveground plant communities. Soil mites can influence the decomposition process by fragmenting apoplankton [13] or by regulating microorganisms (primary decomposers) and protozoa (secondary decomposers) [14], thus enhancing the mineralization rate and inorganic nutrient supply of soil nutrients to promote plant growth [15] and playing an essential role in regulating the species composition, structure, and succession of aboveground plant communities [16,17]. At the same time, changes in forest aboveground plant communities directly affect the composition and nutrient content of forest litter, altering the quality and quantity of food for soil mites, which affects the composition of belowground soil mite communities [18]. However, in the unique karst forest ecosystem of the Heritage Property, there is a lack of comprehensive knowledge on whether differences in vegetation type affect soil mite community structure and whether there are differences in how different vegetation types are affected.

Based on this, the research takes four types of karst forest vegetation under the vertical elevation gradient of the Heritage Property as the research object, and explores the differences and change characteristics of soil mite community structures under different vertical vegetation types of the Heritage Property through a sampling survey and comparative analysis, to provide relative soil biological data information for the health assessment and scientific conservation of the forest ecosystem of the Heritage Property.

## 2. Materials and Methods

### 2.1. Overview of the Study Area

The Shibing Karst World Natural Heritage Property is located in Shibing County, Qiandongnan, Guizhou Province, on the pre-hill slope of the transition zone from the Yunnan-Guizhou Plateau to the low-hill region of Hunan Province. It is also the boundary of the transition zone of the topographic gradient between the second and third terraces in China, overlapping with the scenic area of the Yuntai Mountain-Shanmu River scenic area-ecological water connotation area, with a total area of 282.95 km$^2$, of which the core area is 102.80 km$^2$. The buffer zone area is 180.15 km$^2$ [4]. The Heritage Property's topography

is fragmented, with karst landscapes developing, and the soil-forming parent rock is Cambrian pure dolomite. The soil is mainly composed of thin layers of limestone weathered from dolomite. The area belongs to the central subtropical monsoonal humid climate zone, with a mild climate and abundant rainfall, with an average annual temperature of about 16 °C and an average annual rainfall of 1220 mm [3]. The area has unique physical and geographical conditions and diverse habitat types, which not only nurtures a diverse number of species, but also preserves large areas of forest vegetation with strong native and relict characteristics and various ecosystem types. This makes it one such area that has relatively well-preserved vegetation, is complex, and has various species in the subtropical karst forest system of southern China; it is also one of the more intact and representative dolomite karst regions left behind. It is also one of the better preserved and more complex and diverse species areas in southern China's subtropical dolomite karst forest system [19].

## 2.2. Research Methodology

### 2.2.1. Sample Selection and Settings

This study uses the Shibing Karst World Natural Heritage Property as the research area, covering the three main core areas of the Yuntai Mountain Scenic Area, Hei Chong, and the lower reaches of the Shanmu River, with the habitat spanning from the river valley to the mountain top, with the lowest elevation being 549 m, and the highest elevation being 1021 m. Based on the elevation topography and vegetation composition, four typical vegetation types were selected: mixed coniferous broad-leaved forest (CBF), evergreen broad-leaved forest (EBF), deciduous broad-leaved forest (DBF), and riverbank scrub (RBS), each with three replications, for a total of 12 plots. Details of each sample site are shown in Table 1.

**Table 1.** Natural environmental characteristics of the study area.

| Forest Type | Altitude | Slope | Latitude and Longitude | Main Plants |
|---|---|---|---|---|
| CBF1 | 991 m | Hilltop | 27°9′4.08″ N, 108°6′56.61″ E | **Tree layer:** *Pinus massoniana, Pinus taiwanensis, Cupressus funebris*, etc., the main dominant species are *Pinus* |
| CBF2 | 1021 m | Hilltop | 27°9′47.57″ N, 108°7′41.20″ E | *massoniana* and *Pinus tai-wanensis*; **Shrub layer:** *Coti-nus coggygria, Quercus phillyreoides, Cam-ellia oieifera,* |
| CBF3 | 952 m | Hilltop | 27°6′34.76″ N, 108°6′26.95″ E | *Myrsine africana*, etc.; **Herb layer:** *Carex, Miscanthus sin-ensis.* |
| EBF1 | 912 m | Mid-Upper Slope | 27°6′34.99″ N, 108°6′18.07″ E | **Tree layer:** *Lindera communis, Pinus taiwanensis, Eurya japonica, Loro-petalum chine-nse*, etc., the main |
| EBF2 | 925 m | Mid-Upper Slope | 27°6′40.18″ N, 108°6′28.86″ E | dominant species are *Lindera com-munis* and *Pinus taiwanensis*; **Shrub layer:** *Lindera communis, Querc-us* |
| EBF3 | 940 m | Mid-Upper Slope | 27°6′29.20″ N, 108°6′30.29″ E | *phillyreoides, Ligustrum lucidum, Rhodode-ndron simsii*, etc.; **Herb layer:** *Iris tectorum, Ophiopogon bodinieri*, etc. |
| DBF1 | 890 m | Lower Middle Slope | 27°6′29.20″ N, 108°16′30.29″ E | **Tree layer:** *Cunninghamia lanceolata, Pittosporum tobira, Cyclobalanops-is glauca*, the main dominant species is |
| DBF2 | 939 m | Lower Middle Slope | 27°6′28.90″ N, 108°6′20.01″ E | *Cunninghamia lanceolata.* etc.; **Shrub layer:** *Lindera gla-uca, Machilus microcarpa Hemsl, Michelia martinii*, etc.; |
| DBF3 | 911 m | Lower Middle Slope | 27°6′34.85″ N, 108°6′30.60″ E | **Herb layer:** *Selaginella tam-ariscina, Ophiorrhiza mungos, Pilea notata*, etc. |
| RBS1 | 560 m | River Valley | 27°4′45.28″ N, 108°4′47.86″ E | **Tree layer:** *Dendrobenthamia angustata, Sloanea sinensis*, etc.; **Shrub layer:** *Distyl-ium dunnianum, Boehmeria* |
| RBS2 | 549 m | River Valley | 27°4′50.18″ N, 108°4′42.21″ E | *nivea, Boehmeria penduliflora*, Rubus corchorifolius, etc., the main dominant species is *Distylium dunnianum.*; |
| RBS3 | 584 m | River Valley | 27°4′44.34″ N, 108°4′41.34″ E | **Herb layer:** *Ficus tikoua, Pogonatherum crinitum, Pronep-hrium gymnopteridifrons, Selaginella uncinata*, etc. |

### 2.2.2. Sample Collection and Processing

In September 2021, six 150 mm × 150 mm sampling points were set up in each of the above sample plots, with each sample point spaced about 10 m apart, and soil samples were collected in each plot according to the s-or snake-type sampling method. After removing the dead leaves from the surface, the 100 mm (D) × 64 mm (H) cylindrical swivel knife was used for sampling. Samples were taken in three layers: humus layer, 0–50 mm soil layer (upper layer), and 50–100 mm soil layer (lower layer). Two hundred and sixteen samples were taken in total (4 habitat types × 3 sets of replicates × 6 sampling points × 3 soil layers). All samples were packed into well-ventilated cloth bags, brought back to the room, and then continuously baked for 48 h using the Tullgren method. The isolated mites were kept in 75% alcohol for cleaning and fixation and were finally preserved and made transparent at room temperature in plastic test tubes containing lactic acid solution. After transparency, the soil mites were made into temporary specimens and placed under a microscope (Olympus CX41RF) for observation. The specimens were morphologically identified in accordance with *the Pictorial Keys to Soil Animals of China* [20], *Acarology* [21], *A manual of Acarology (3rd edition)* [22], and *Soil Gamasid Mites in Northeast China* [23]. All samples were identified to the genus. The classification order mainly adopts the classification system of *A manual of Acarology (3rd edition)* [22].

2.2.3. Data Calculation and Analysis

(1)  Community dominance [24,25]: more than 10% of the total catch in the number of individuals is the dominant group (++++), 1~10% of the total catch in the number of individuals is the common group (+++), 0.5~1% of the total catch in the number of individuals is the rare group (++), and less than 0.5% of the total catch in the number of individuals is the most rare group (+).

(2)  Community structure analysis [26]: the Shannon–Winner diversity index (*H*), Margalef richness index (*SR*), and Pielou evenness index (*J*) were used to characterize the community jointly.

$$H = -\sum_{i=1}^{s} P_i \ln P_i \tag{1}$$

where: $P_i$ is the ratio of the number of individuals of genus *i* to the number of all individuals.

$$SR = (S-1)/\ln N \tag{2}$$

where: *N* is the total number of individuals of all groups in the soil mite community.

$$J = H/\ln S \tag{3}$$

where: *H* is the Shannon–Winner diversity index and *S* is the number of groups (genera).

(3)  Analysis of community similarity and variability [27] using Jaccard's similarity coefficient:

$$q = c/(a+b-c) \tag{4}$$

where: *a* is the number of groups in community A, *b* is the number of groups in community B, and *c* is the number of groups common to both communities.

The Marczewski–Steinhaus distance (Cms) [8,28] was used for cluster analysis (average clustering method) with Origin software for statistical computing and graphics. The original count matrix was converted to a binary matrix when Cms was used for cluster analysis. Cms is, in fact, complementary to the Jaccard similarity index, Cms = $1 - C_J$.

(4)  *MI* analysis of the maturity of the predatory gamasid mites (Mesostigmata: Gamasina) [29,30].

$$MI = \sum_{i=1}^{s} K_i / (\sum_{i=1}^{s} K_i + \sum_{i=1}^{s} r_i) \tag{5}$$

where: *s* is the genera of gamasid mites; $K_i$ is the *K* value of the family where the genus *i* belongs, and $r_i$ is the *r* value of the family where the genus *i* belongs. *MI* < 0.5, *r* selected; *MI* = 0.5, *K* and *r* selected; *MI* > 0.5, *K* selected.

(5)  MGP analysis of the ecological structure of oribatid mites (Acari:Oribatid): The ecological groups of oribatid mites were analyzed using the MGP analysis of mites [21,31,32], M-Macropylina, G-Gymnonota, and P-Poronota. The percentage of genera of each group was calculated for MGP I analysis and the percentage of individuals of each group was calculated for MGP II analysis, respectively. The criteria for community type classification are shown in Table 2.

**Table 2.** Measurable standard of community types on oribatid mites (Acari:Oribatid).

| Community Types | Abbreviation | Value Ranges of Mites (Oribatida) Group |
|---|---|---|
| Macropylina type | M | M > 50% |
| Gymnonota type | G | G > 50% |
| Poronota type | P | P > 50% |
| Overall type | O | 20% < M, G, P < 50% |
| Macropylina-Gymnonota type | MG | M, G = 20~50%, P < 20% |
| Gymnonota-Poronota type | GP | G, P = 20~50%, M < 21% |
| Macropylina-Poronota type | MP | M, P = 20~50%, G < 22% |

(6)  Statistical analysis: The data were statistically organized using Microsoft Excel 2020 statistical software; IBM SPSS 22.0 was used to analyze the data, and for data obeying normal distribution, one-way ANOVA was used to compare the differences in the number of soil mite genera, number of individuals, diversity index, richness index, evenness index, etc., under the vertical zone spectrum of vegetation in the heritage sites. The LSD test was used for the significance of the differences between sites (significant difference: $p < 0.05$); log(x + 1) transformation was performed for the data that did not obey normal distribution, and if they still did not follow the normal distribution, the test was performed with non-parametrics [33,34]. The above analysis and graphing were completed using Origin 2021.0 and IBM SPSS 22.0.

## 3. Results and Analysis

### 3.1. Soil Mite Community Composition and Dominance

3.1.1. General Composition and Dominance of Soil Mite Communities

A total of 10,563 soil mites were captured in the study area, belonging to 3 orders, 67 families, and 137 genera. Overall, the composition of the soil mites by the order was 1758 individuals from 20 families and 44 genera of Mesostigmata, 70 individuals from 5 families and 7 genera of Trombidiformes, and 8735 individuals from 42 families and 86 genera of Sarcoptiformes (Table 3). Among them, the percentage of families to the total number of families, the ratio of genera to the total number of genera, and the number of individuals to the total number of individuals were all different for Sarcoptiformes (62.69%, 62.77%, and 82.69%). The differences in the number of soil mite families as a percentage of the total number of families, the number of genera as a percentage of the total number of genera, and the number of individuals as a percentage of the total number of individuals all showed that Sarcoptiformes (62.69%, 62.77%, and 82.69%) > Mesostigmata (29.85%, 32.12%, and 16.64%) > Trombidiformes (7.46%, 5.11%, and 0.66%). It can be seen that the family and genus composition of Sarcoptiformes–Oribatida dominates the soil mites in the study area, and oribatid mites constitute the basic components of soil mites in the karst forests of the Heritage Property.

In terms of the composition of the various genera of soil mites, *Perscheloribates* and *Scheloribates* are the dominant groups, 20 genera such as *Pachylaelaps*, *Epilohmannia*, and *Nothrus* are common groups, *Cosmolaelaps*, *Hypochthonius* and *Lasiobelba*, and 15 other genera are rare groups, and 100 genera including *Podocinum*, *Incabates*, and *Perxylobates* are the most rare groups. The percentages of dominant, common, rare, and most rare groups in the total number of soil mite genera were 1.46%, 14.6%, 10.95%, and 72.99%, respectively. The percentages of the total number of individuals in soil mites were 24.09%, 55.54%, 10.07%, and 10.29%, respectively. It is clear that the number of soil mite genera in the karst forests of the Heritage Property is dominated by most rare groups, and the number of individuals is dominated by common groups.

**Table 3.** Composition and population distribution of soil mite communities in the study area.

| Family | Genus | CBF | EBF | DBF | RBS | Total |
|---|---|---|---|---|---|---|
| Mesostigmata (Order) | | | | | | |
| Trachytidae | *Trachytes* | 1 (+) | 6 (+) | 3 (+) | | 10 (+) |
| | *Uropoda* | 2 (+) | | 1 (+) | | 3 (+) |
| | *Discourella* | 2 (+) | | | 1 (+) | 3 (+) |
| Trematuridae | *Nenteria* | 66 (+++) | 17 (+++) | 64 (+++) | 5 (+) | 152 (+++) |
| Dinychidae | *Dinychus* | 5 (+) | | 9 (+) | | 14 (+) |
| | *Uroobovella* | 9 (+) | 1 (+) | 15 (++) | 2 (+) | 27 (+) |
| Oplitidae | *Oplitis* | | 2 (+) | 9 (+) | 5 (+) | 16 (+) |
| | *Epicrius* | | 2 (+) | | | 2 (+) |
| Zerconidae | *Metazercon* | | | 3 (+) | | 3 (+) |
| | *Parazercon* | 4 (+) | | 2 (+) | | 6 (+) |
| | *Prozercon* | 2 (+) | | | | 2 (+) |
| | *Xenozercon* | | | | 2 (+) | 2 (+) |
| | *Syskenozercon* | | | 1 (+) | | 1 (+) |
| | *zercon* | | | 5 (+) | 2 (+) | 7 (+) |
| Parasitidae | *Neogamasus* | 18 (++) | 23 (+++) | 65 (+++) | 26 (++) | 132 (+++) |
| | *Vulgarogamasus* | 8 (+) | | 6 (+) | 3 (+) | 17 (+) |
| | *Cornigamasus* | 1 (+) | | | | 1 (+) |
| | *Parasitus* | 116 (+++) | 47 (+++) | 50 (+++) | 24 (++) | 237 (+++) |
| Veigaiidae | *Veigaia* | 2 (+) | 2 (+) | 4 (+) | | 8 (+) |
| Rhodacaridae | *Gamasellus* | 3 (+) | | | | 3 (+) |
| | *Dendrolaelaps* | 4 (+) | 12 (++) | 17 (++) | 2 (+) | 35 (+) |
| | *Rhodacarus* | 5 (+) | 5 (+) | 97 (+++) | 21 (++) | 128 (+++) |
| | *Rhodacarellus* | 1 (+) | | | | 1 (+) |
| Ologamasidae | *Gamasiphis* | | 1 (+) | 2 (+) | | 3 (+) |
| Macrochelidae | *Glyptholaspis* | 2 (+) | 13 (++) | | | 15 (+) |
| | *Macrocheles* | 26 (+++) | 20 (+++) | 20 (++) | 2 (+) | 68 (++) |
| Parholaspididae | *Gamasholaspis* | | 2 (+) | 8 (+) | | 10 (+) |
| | *Krantzholaspis* | 6 (+) | 14 (++) | | | 20 (+) |
| | *Parholaspulus* | 71 (+++) | 67 (+++) | 75 (+++) | 34 (++) | 247 (+++) |
| Pachylaelapidae | *Pachylaelaps* | 29 (+++) | 43 (+++) | 28 (++) | 21 (++) | 121 (+++) |
| | *Pachyseius* | 4 (+) | 4 (+) | 18 (++) | 7 (+) | 33 (+) |
| Phytoseiidae | *Amblyseius* | | | 2 (+) | | 2 (+) |
| Ascidae | *Asca* | 1 (+) | 15 (+++) | | | 16 (+) |
| Ameroseiidae | *Ameroseius* | 3 (+) | 11 (++) | 7 (+) | 4 (+) | 25 (+) |
| Polyaspididae | *Allosuctobelba* | | 1 (+) | | | 1 (+) |

**Table 3.** *Cont.*

| Family | Genus | CBF | EBF | DBF | RBS | Total |
|---|---|---|---|---|---|---|
| Podocinidae | *Podocinum* | | 8 (++) | 11 (+) | | 19 (+) |
| Blattisociidae | *Cheiroseius* | 14 (++) | 10 (++) | 8 (+) | | 32 (+) |
| | *Lasioseius* | 44 (+++) | 6 (+) | 54 (+++) | 76 (+++) | 180 (+++) |
| | *Proctolaelaps* | | | 2 (+) | | 2 (+) |
| Laelapidae | *Cosmolaelaps* | 18 (++) | 18 (+++) | 25 (++) | | 61 (++) |
| | *Geolaelaps* | 27 (+++) | 26 (+++) | 23 (++) | 8 (+) | 84 (++) |
| | *Laelaspis* | | 1 (+) | 1 (+) | | 2 (+) |
| | *Ololaelaps* | 1 (+) | | | | 1 (+) |
| | *Alloparasitus* | | 4 (+) | 2 (+) | | 6 (+) |
| Labidostomatidae | *Labidostoma* | | | 6 (+) | 4 (+) | 10 (+) |
| Cunaxidae | *Cunaxa* | | 1 (+) | | | 1 (+) |
| | *Dactyloscirus* | | 1 (+) | | | 1 (+) |
| Trombidiformes (Order) | | | | | | |
| Erythraeidae | *Balaustium* | 6 (+) | | | | 6 (+) |
| Microtrombidiidae | *Echinothrombium* | | | 4 (+) | 2 (+) | 6 (+) |
| | *Microtrombidium* | 6 (+) | 4 (+) | 19 (++) | 11 (+) | 40 (+) |
| Stigmaeidae | *Stigmaeus* | 1 (+) | | | 5 (+) | 6 (+) |
| Sarcoptiformes(Order) | | | | | | |
| Mesoplophoridae | *Archoplophora* | 96 (+++) | 75 (+++) | 10 (+) | 12 (+) | 193 (+++) |
| Hypochthoniidae | *Nehypochthon* | 7 (+) | | | | 7 (+) |
| | *Eohypochthonius* | | 15 (+++) | | | 15 (+) |
| | *Hypochthonius* | | | 51 (+++) | 4 (+) | 55 (++) |
| | *Hypochthoniella* | 2 (+) | | | | 2 (+) |
| Lohmanniidae | *Meristacarus* | | 4 (+) | | | 4 (+) |
| | *Mixacarus* | | 7 (+) | | 1 (+) | 8 (+) |
| | *Papillacarus* | | 40 (+++) | | | 40 (+) |
| | *Lohmannia* | | 4 (+) | | | 4 (+) |
| Eulohmaimiidae | *Eulohmannia* | 1 (+) | | | | 1 (+) |
| Epilohmanniidae | *Epilohmannia* | 123 (+++) | 7 (+) | 206 (+++) | | 336 (+++) |
| Euphtliiracaiidae | *Acrotritia* | | 3 (+) | | | 3 (+) |
| | *Microtritia* | 3 (+) | 1 (+) | | | 4 (+) |
| | *Rhysotritia* | 183 (+++) | 74 (+++) | 99 (+++) | 55 (+++) | 411 (+++) |
| Synichotritiidae | *Synichotritia* | | | 1 (+) | | 1 (+) |
| Phthiracaridae | *Steganacarus* | 20 (++) | 2 (+) | | | 22 (+) |
| | *Hoplophthiracarus* | 8 (+) | 1 (+) | 2 (+) | | 11 (+) |

**Table 3.** *Cont.*

| Family | Genus | CBF | EBF | DBF | RBS | Total |
|---|---|---|---|---|---|---|
| | *Phthiracarus* | 2 (+) | 5 (+) | | | 7 (+) |
| Camisiidae | *Camisia* | 42 (+++) | | 15 (++) | 12 (+) | 69 (++) |
| | *Platynothrus* | 2 (+) | | | 1 (+) | 3 (+) |
| | *Heminothrus* | | | 33 (+++) | 26 (++) | 59 (++) |
| Nothridae | *Nothrus* | 100 (+++) | 11 (++) | 57 (+++) | 169 (+++) | 337 (+++) |
| Crotonioidea | *Crotonia* | | | 20 (++) | | 20 (+) |
| | *Holonothrus* | | | 2 (+) | | 2 (+) |
| Trhypochthoniidae | *Archegozetes* | 3 (+) | | | 17 (+) | 20 (+) |
| | *Allonothrus* | 1 (+) | | | 61 (+++) | 62 (++) |
| | *Afronothrus* | 10 (+) | | 1 (+) | 3 (+) | 14 (+) |
| | *Trhypochthoniellus* | 1 (+) | | | | 1 (+) |
| | *Trhypochthonius* | 8 (+) | 19 (+++) | 1 (+) | 3 (+) | 31 (+) |
| Malaconothridae | *Malaconothrus* | 1 (+) | 4 (+) | 1 (+) | | 6 (+) |
| | *Trimalaconothrus* | 1 (+) | | 3 (+) | | 4 (+) |
| Nanhermanniidae | *Crythermannia* | 14 (++) | 6 (+) | 1 (+) | | 21 (+) |
| | *Nanhermannia* | 2 (+) | 16 (+++) | | | 18 (+) |
| Hermanniidae | *Hermannia* | 9 (+) | 3 (+) | 3 (+) | | 15 (+) |
| Hennaiiniellidae | *Hermanniella* | 1 (+) | 6 (+) | | | 7 (+) |
| Plasmobatidae | *Plasmobates* | | 2 (+) | | | 2 (+) |
| Damaeidae | *Damaeus* | | | 1 (+) | | 1 (+) |
| | *Epidamaeus* | 14 (++) | 2 (+) | 5 (+) | 6 (+) | 27 (+) |
| Eremulidae | *Eremulus* | | | 14 (+) | 30 (++) | 44 (+) |
| Eremobelbidae | *Eremobelba* | 43 (+++) | 2 (+) | 20 (++) | 17 (+) | 82 (++) |
| Liacaridae | *Liacarus* | 1 (+) | 3 (+) | | 6 (+) | 10 (+) |
| Carabodidae | *Carabodes* | | 1 (+) | | | 1 (+) |
| Otocepheidae | *Dolicheremaeus* | | 1 (+) | | | 1 (+) |
| Eremaeidae | *Eremaeus* | 4 (+) | 6 (+) | | | 10 (+) |
| Ctenobelbidae | *Ctenobelba* | | | 6 (+) | | 6 (+) |
| Peloppiidae | *Ceratoppia* | 17 (++) | 5 (+) | 1 (+) | 41 (+++) | 64 (++) |
| | *Austroceratoppia* | 6 (+) | | | | 6 (+) |
| | *pyroppia* | | 1 (+) | | 1 (+) | 2 (+) |
| Astegistidae | Cultroribula | 42 (+++) | 14 (++) | | 489 (++++) | 545 (+++) |
| Gustaviidae | *Gustavia* | 19 (++) | 1 (+) | | 15 (+) | 35 (+) |
| Suctobelbidae | *Polyaspinus* | 4 (+) | | | | 4 (+) |
| | *Suctobelba* | 26 (+++) | 8 (++) | 2 (+) | | 36 (+) |
| | *Parisuctobelba* | 1 (+) | | | | 1 (+) |

**Table 3.** *Cont.*

| Family | Genus | CBF | EBF | DBF | RBS | Total |
|---|---|---|---|---|---|---|
| Oppiidae | *Lasiobelba* | 16 (++) | | 8 (+) | 60 (+++) | 84 (++) |
| | *Lauroppia* | | | 1 (+) | 3 (+) | 4 (+) |
| | *Medioppia* | 1 (+) | | 2 (+) | | 3 (+) |
| | *Oppia* | 2 (+) | 3 (+) | 2 (+) | 16 (+) | 23 (+) |
| | *Oppiella* | 19 (++) | 32 (+++) | 36 (+++) | 17 (+) | 104 (++) |
| | *Ramusella* | 1 (+) | 17 (+++) | 3 (+) | | 21 (+) |
| | *Phauloppia* | | 2 (+) | | | 2 (+) |
| | *Arcoppia* | | 11 (++) | | | 11 (+) |
| | *Microppia* | | 5 (+) | | | 5 (+) |
| Nippobodidae | *Nippobodes* | 2 (+) | | 4 (+) | | 6 (+) |
| Tectocepheidae | *Tectocepheus* | 3 (+) | 83 (+++) | | 338 (+++) | 424 (+++) |
| Scutoverticidae | *Scutovertex* | 42 (+++) | 7 (+) | 3 (+) | 1 (+) | 53 (++) |
| Parakalumniidae | *Neoribates* | 5 (+) | 14 (++) | 4 (+) | 41 (+++) | 64 (++) |
| Scheloribatidae | *Perscheloribates* | 260 (++++) | 113 (+++) | 499 (++++) | 561 (++++) | 1433 (++++) |
| | *Scheloribates* | 78 (+++) | 41 (+++) | 181 (+++) | 812 (++++) | 1112 (++++) |
| Mochlozetidae | *Mochlozetes* | | 1 (+) | | 6 (+) | 7 (+) |
| Haplozetidae | *Incabates* | | 11 (++) | | | 11 (+) |
| | *Paraxylobates* | | 1 (+) | | | 1 (+) |
| | *Peloribates* | 141 (+++) | 56 (+++) | 166 (+++) | 62 (+++) | 425 (+++) |
| | *Perxylobates* | | 2 (+) | | | 2 (+) |
| | *Rostrozetes* | 50 (+++) | 6 (+) | 51 (+++) | | 107 (+++) |
| | *Setoxylobates* | 10 (+) | 28 (+++) | 3 (+) | 96 (+++) | 137 (+++) |
| | *Vilhenabates* | 151 (+++) | 29 (+++) | 31 (+++) | 296 (+++) | 507 (+++) |
| Xylobatidae | *Xylobates* | 57 (+++) | 21 (+++) | 179 (+++) | 230 (+++) | 487 (+++) |
| Protoribatidae | *Protoribates* | | 22 (+++) | | | 22 (+) |
| Ceratozetidae | *Ceratozetes* | 19 (++) | 44 (+++) | | 30 (++) | 93 (++) |
| Galumnidae | *Galumna* | 22 (++) | 17 (+++) | 23 (++) | | 62 (++) |
| | *Pergalumna* | 6 (+) | 14 (++) | | | 20 (+) |
| | *Protokalumna* | 3 (+) | 13 (++) | 5 (+) | | 21 (+) |
| | *Trichogalumna* | 94 (+++) | 101 (+++) | 408 (++++) | 31 (++) | 634 (+++) |
| | *Porogalumnella* | | | 125 (+++) | 2 (+) | 127 (+++) |
| | *Galumnella* | | | 5 (+) | | 5 (+) |
| Histiostomatidae | *Histiotoma* | 7 (+) | 1 (+) | 20 (++) | | 28 (+) |
| Total number of taxa | | 90 | 90 | 81 | 59 | 320 |
| Total number of individuals | | 2314 | 1431 | 2980 | 3838 | 10,563 |

Note: ++++: Dominant group (>10%); +++: Common group (1–10%); ++: Rare group (0.5–1%); +: Most rare group (<0.5%).

3.1.2. Community Composition and Dominance of Soil Mites under Different Vegetation Types

From the composition of soil mites under each vegetation type. A total of 2314 soil mites belonging to 49 families and 90 genera were captured in the CBF. One thousand, four hundred and thirty-one soil mites belonging to 47 families and 90 genera were captured in the EBF. One thousand, nine hundred and eighty soil mites belonging to 43 families and 81 genera were captured in the DBF. Three thousand, eight hundred and thirty-eight soil mites belonging to 38 families and 59 genera were captured in the RBS. In descending order, the number of soil mite families and genera were CBF > EBF > DBF > RBS. The variation in the number of individual mites was RBS > DBF > CBF > EBF (Figure 1A). The one-way ANOVAs for the number of soil mite genera and individuals under different vegetation types are shown in Table 4. The results show that there exists a significant difference in the number of soil mite genera between RBS and the other three vegetation types ($p < 0.05$), and there seems to be a significant difference in the number of soil mite individuals between RBS and EBF ($p < 0.05$), but no significant difference between the other vegetation types. In summary, the three vegetation types CBF, EBF, and DBF are rich in the number of soil mite families and genera in the Heritage property karst forest ecosystem, whereas the RBS habitat is richer in the number of soil mite individuals, which may be related to the number of individuals of the dominant soil mite groups in the RBS habitat.

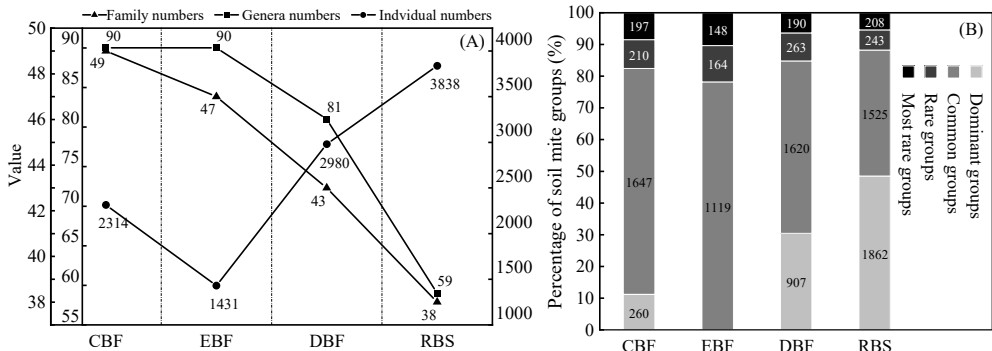

**Figure 1.** Variation in the number of families, genera, and individuals of soil mites (**A**) and percentage variation in soil mite community dominance (**B**) under different vegetation types in the study area.

**Table 4.** Distributions of genera numbers and individual numbers of soil mites (Mean $\pm$ SE, $n = 3$).

| Community Parameter | CBF | EBF | DBF | RBF | Mean | P |
|---|---|---|---|---|---|---|
| Genus numbers | 56.33 $\pm$ 1.76 a | 56 $\pm$ 5.03 a | 51.67 $\pm$ 6.74 a | 34.67 $\pm$ 5.24 b | 49.67 $\pm$ 3.43 | 0.048 |
| Individual numbers | 540.33 $\pm$ 80.77 ab | 376.33 $\pm$ 28.62 b | 725.67 $\pm$ 186.11 ab | 856 $\pm$ 134.56 a | 624.58 $\pm$ 75.78 | 0.099 |

Note: Different lowercase letters indicate significant differences in the number of soil mite genera (number of individuals) between the different vegetation types ($p < 0.05$).

In terms of the dominance of soil mites in different vegetation types, *Perscheloribates* was the dominant group in CBF habitats, accounting for 11.2% of the total catch; common, rare, and most rare groups accounted for 71.2%, 9.1%, and 8.5% of the total catch, respectively. The common, rare, and most rare groups accounted for 78.2%, 11.5%, and 10.3% of the total catch. *Perscheloribates* and *Trichogalumna* were the dominant groups in the DBF, accounting for 30.4% of the total catch; common, rare, and most rare groups accounted for 54.4%, 8.8%, and 6.4% of the total catch, respectively. The dominant groups in the RBS were *Cultroribula*, *Perscheloribates*, and *Scheloribates*, which accounted for 48.5% of the total catch, whereas the common, rare, and most rare groups accounted for 39.7%, 6.3%, and 5.4% of the total catch, respectively (Figure 1B). In summary, the dominant groups of soil mites were not entirely consistent across vegetation types in the Heritage property; the composition of the dominant and common groups of soil mites varied considerably in



terms of the number of individuals, with the common group of soil mites dominating the number of individuals in each habitat except for RBS.

### 3.2. Soil Mites Community Structure

#### 3.2.1. Horizontal and Vertical Distribution of the Number of Genera and Individuals of Soil Mites

From the horizontal structure of soil mite community distribution (Figure 2A), the number of soil mite genera in the humic layer was EBF > CBF > DBF > RBS, among which RBS was significantly different from the other three vegetation zones ($p < 0.05$); the variation in the number of individuals was characterized as RBS > DBF > CBF > EBF, among which EBF was significantly different from RBS ($p < 0.05$). The number of soil mite genera and individuals in the soil upper layer and soil lower layer did not differ significantly ($p > 0.05$) between the four vegetation zones, with the soil upper layer in CBF having the highest number of soil mite genera; the number of mite individuals in soil upper layer differed significantly ($p < 0.05$) between the four vegetation zones, with RBS habitat having the highest number of soil mite individuals and EBF habitat having the lowest.

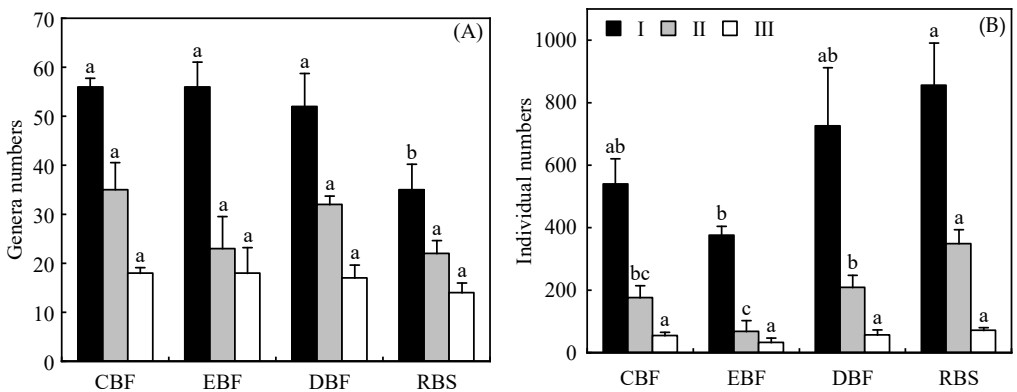

**Figure 2.** Horizontal and vertical varieties on genera numbers (**A**) and individual numbers (**B**) of soil mites in the study area. I: Humic layer; II: Soil upper layer (0–50 mm); III: Soil lower layer (50–100 mm). The same below. Note: Different lowercase letters indicate significant differences in the number of soil mite genera (number of individuals) between different vegetation types under the same soil layer ($p < 0.05$).

In terms of vertical distribution (Figure 2B), the variation in the number of soil mite genera and individuals in the four typical forest communities under the vertical gradient in the elevation of the Heritage Property showed that the humic layer > soil upper layer > soil lower layer. The soil mites in the humic layer were dominant, i.e., the number of soil mite genera and individuals decreased with increasing soil depth, showing prominent surface aggregation characteristics. The surface aggregation of the number of soil mite individuals was more apparent than the number of soil mite genera.

#### 3.2.2. Community Diversity

The variation of the soil mite communities Shannon–Winner (*H*), Margalef (*SR*), and Pielou (*J*) under the different vegetation types is shown in the Figure looking at the values of each index at the level of the four vegetation types (Figure 3A). EBF has the highest diversity index, richness index, and evenness index, and RBS has the lowest values of diversity index and richness index, where there is a significant difference between EBF and RBS in diversity index ($p < 0.05$) and richness index between vegetation types ($p < 0.05$).

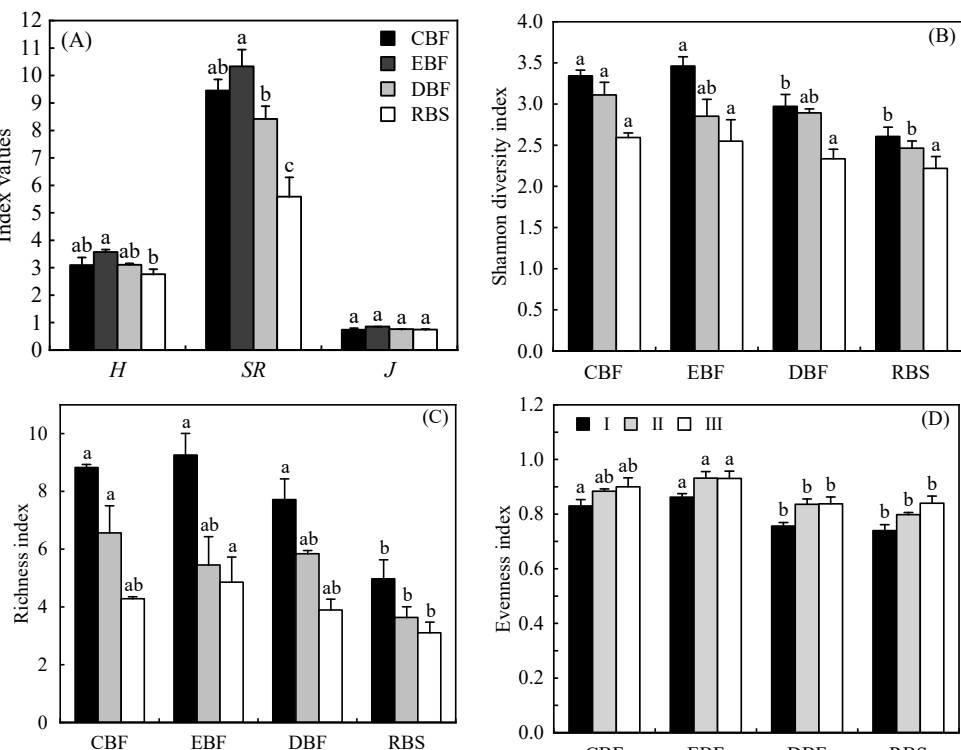

**Figure 3.** The horizontal (**A**) and vertical distribution (**B**–**D**) of the Shannon diversity index (*H*), Margalef richness index (*SR*), and Pielou evenness index (*J*) of soil mites in the study area. Note: Different lowercase letters indicate significant differences between vegetation types for each parameter of community diversity ($p < 0.05$).

From the values of the indices on the vertical soil layers of the four vegetation types, the diversity and richness indices show a gradual decrease with the deepening of the soil layer (Figure 3B,C), whereas the uniformity index generally indicates a pattern of variation in which the soil layer is larger than the humic layer (Figure 3D).

Both the diversity and richness indices of the humic layer showed EBF > CBF > DBF > RBS, whereas the evenness index was highest in the EBF habitat, with slight variation between DBF and RBS, where each index of RBS was significantly different ($p < 0.05$) from CBF and EBF, respectively, among the vegetation types.

The diversity and richness indices of the soil upper layer both showed CBF > DBF > EBF > RBS, with a significant difference between CBF and RBS ($p < 0.05$); the evenness index showed EBF > CBF > DBF > RBS, with a significant difference between EBF and DBF and RBS ($p < 0.05$).

The diversity index of the soil upper layer was characterized as CBF > EBF > DBF > RBS, with no significant difference between the four vegetation types ($p > 0.05$); both the richness index and evenness index were highest in the EBF habitat. There is a significant difference between EBF and RBS in the richness index ($p < 0.05$), and between EBF and DBF and RBS in the evenness index ($p < 0.05$), respectively.

### 3.2.3. Community Similarities and Differences

The results of the similarity analysis of soil mite species in the four typical forest communities under the vertical zone spectrum of vegetation in the Heritage Property (Table 5) showed that the similarity index between communities in this region was medium dissimilar, except for between CBF and DBF, which was medium similar, and were generally not very similar.

**Table 5.** Distributions of genera numbers and individual numbers of soil mites in the study area.

| Forest Type | CBF | EBF | DBF | RBS |
|---|---|---|---|---|
| CBF | 1 | 0.4173 | 0.5 | 0.4466 |
| EBF | | 1 | 0.4492 | 0.3796 |
| DBF | | | 1 | 0.4737 |
| RBS | | | | 1 |

0 < q < 0.25: Extremely dissimilar; 0.25 ≤ q < 0.5: Medium dissimilar; 0.5 ≤ q < 0.75: Medium similar; 0.75 ≤ q < 1: Extremely similar.

To further analyze the similarities and differences between the communities of each site, the top ten soil mites with the highest percentage of individuals in the 12 sample sites under the four vegetation vertical zones were selected for two-way cluster analysis. The results are shown in Figure 4. Based on the Marczewski–Steinhaus distance (Cms), the soil mite community structure in the four vegetation zones was classified into seven types. Type 1: EBF 1; Type 2: CBF 2, CBF 3; Type 3: EBF 1, EBF 3; Type 4: EBF 2; Type 5: DBF 1; Type 6: DBF 2, DBF 3; Type 7: RBS 1, RBS 2, RBS 3.

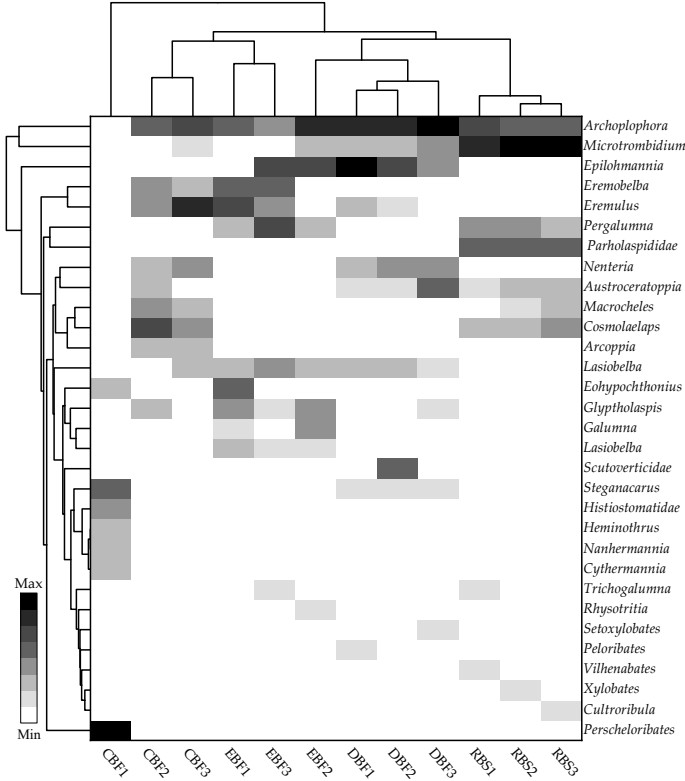

**Figure 4.** Two-way cluster analysis of the soil mites' communities under different vegetation vertical zones in in the study area.

### 3.3. Ecological Group of Predatory Gamasid Mites (Mesostigmata: Gamasina)

The determination of *K*- and *r*-values for the predatory gamasid mites was based mainly on the daily egg production rate and development rate of each family of mites and the dispersal ability and population dynamics of the mites. Refer to the relevant references for specific assignment criteria [29,30,35] (Table 6). The results of the maturity index MI calculation for the predatory gamasid mites in each habitat are shown in Table 7. The results showed that the predatory gamasid mites in the study area consisted of 14 families, and in general, the four habitats were dominated mainly by *r*-selected species, among which the *K*-selected and *r*-selected groups of RBS were comparable; from different soil layers, both humic and soil layers were dominated by *r*-selected species. It can be seen that the predatory gamasid mites in the study area are mainly groups that adopt reproductive

responses, and the differences in the ecological groups of the predatory gamasid mites between vegetation types are not significant.

**Table 6.** *K*- or *r*-values of predatory soil mites (Mesostigmata: Gamasina) in the study area.

| Family | *K* Value | Family | *r* Value |
|---|---|---|---|
| Pachylaelapidae | 1 | Laelapidae | 1 |
| Parholaspididae | 2 | Ologamasidae | 1 |
| Veigaiidae | 2 | Podocinidae | 1 |
| Rhodacaridae | 2 | Ascidae | 2 |
| Epicriidae | 3 | Blattisociidae | 2 |
| Ameroseiidae | 3 | Phytoseiidae | 2 |
| Zerconidae | 3 | Parasitidae | 4 |

**Table 7.** *MI* values of predatory soil mites (Mesostigmata: Gamasina) in the study area.

| Sampling Layer | CBF | | EBF | | DBF | | RBS | |
|---|---|---|---|---|---|---|---|---|
| | *MI* | Group | *MI* | Group | *MI* | Group | *MI* | Group |
| L | 0.4651 | *r* | 0.4242 | *r* | 0.4222 | r | 0.5 | *K/r* |
| S | 0.4242 | *r* | 0.3846 | *r* | 0.4286 | r | 0.4762 | *r* |
| Total | 0.4894 | *r* | 0.4865 | *r* | 0.4902 | r | 0.5 | *K/r* |

### 3.4. Ecological Group of Oribatid Mites (Acari:Oribatid)

The MGP analysis of soil mite community structure under different vegetation types is shown in Table 8. In terms of the percentage of genera, all four habitats were O-types, with a more significant proportion of M- and G-groups in CBF, DBF, and RBS, and a more substantial proportion of G- and P-groups in EBF; in terms of the percentage of individuals, all four habitats were P-types.

**Table 8.** Community structure of soil mites (Acari: Oribatid) at four typical vegetation in the study area.

| Forest Type | Genera Percent (%) | | | Community Types | Individual Percent (%) | | | Community Types |
|---|---|---|---|---|---|---|---|---|
| | Macropyline | Gymnonota | Poronota | | Macropyline | Gymnonota | Poronota | |
| CBF | 35.71 | 39.29 | 25.00 | O | 34.13 | 13.73 | 52.14 | P |
| EBF | 27.59 | 39.66 | 32.76 | O | 26.08 | 22.05 | 51.87 | P |
| DBF | 33.33 | 35.56 | 31.11 | O | 21.88 | 4.75 | 73.37 | P |
| RBS | 32.43 | 35.14 | 32.43 | O | 10.19 | 29.10 | 60.71 | P |

### 3.5. Trophic Structure of Oribatid Mites (Acari:Oribatid)

Oribatid mites are critical decomposers of soil organic matter and play an essential ecological role in soil ecosystems [22,36–38]. The differentiation of trophic ecological niches is thought to explain the coexistence of a large number of soil organism species. Schneider et al. [39] classified soil mites into four major groups in terms of trophic structure, namely:

(1) Carnivorous and omnivorous oribatid mites.
(2) Secondary decomposers.
(3) Primary decomposers.
(4) Phycophagous and fungivorous oribatid mites.

A total of 86 genera of the oribatid mites are distributed under the vertical gradient of the vegetation in the study area (Table 3). The classification results according to the groups mentioned above, taking into account the relevant literature [40–43], are shown in Table 9. The results show that under the vertical gradient of vegetation in the study area, each trophic level has different soil mites, which initially constitute a complete decomposer trophic structure or decomposition functional group of the soil ecosystem functional type in the study area.

**Table 9.** Trophic structure of soil mites (Acari: Oribatida) in the study area.

| Oribatid Mite Groups | Feeding Guild | Food Materials |
|---|---|---|
| *Hypochthonius, Lasiobelba, Lauroppia, Medioppia, Oppia, Oppiella, Ramusell, Phauloppia, Arcoppia, Microppia, Nothrus, Galumna, Pergalumna, Protokalumna, Trichogalumna, Porogalumnella, Galumnella* (17) | Carnivores/Scavengers/Omnivores | Living and dead animals (nematodes, collembolans) and fungi |
| *Perscheloribates, Scheloribates, Crythermannia, Nanhermannia, Malaconothrus, Carabodes* (6) | Secondary decomosers | Predominantly fungi, in part litter |
| *Liacarus, Tectocepheus, Heminothrus* (3) | Primary decomposers | Predominantly litter |
| *Eremobelba, Cultroribula, Ceratozetes, Mochlozetes* (4) | Phycophages/ Fungivores | Lichens and algae |

## 4. Discussion

### 4.1. Differences in Soil Mite Groups in Different Montane Forest Environments

The Shibing Karst World Natural Heritage Property is a typical subtropical mountainous area. The typical subtropical forest located in the Heritage Property has been described as an 'oasis' on the Tropic of Cancer [4]. A total of 67 families and 137 genera of soil mites have been found under the vertical band spectrum of the mountainous area of the Heritage Property. Among them, *Perscheloribates* and *Scheloribates* are the dominant groups in the area. They are abundantly distributed in all four vegetation zones, indicating that these two groups have successfully adapted to the montane forest environment of the Heritage Property and that their dynamics are essential indicators of the montane forest soil environment [10]. In contrast to other subtropical mountains in China, the Fanjing Mountains have *Oppiella, Xylobates*, and *Suctobelbella* as dominant groups and a relatively high proportion of *Scheloribates* [44]; Chishui Alsophila Natural Reserve has *Tectocepheus* as the dominant group [45]; *Oppiella* and *Suctobelbella* are the dominant groups in Huangshan [46]; and *Chamobates* and *Minunthozetes* are the dominant groups in the Mitra National Park [47].

Of the four vegetation types in the study area, the more abundant soil mite groups in the RBS were *Cultroribula, Perscheloribates*, and *Scheloribates*; the more abundant groups in the DBF were *Trichogalumna* and *Perscheloribates*; the more abundant groups in the EBF were *Perscheloribates* and *Archoplophora*; the more numerous groups in the CBF are *Perscheloribates* and *Vilhenabates*. Combined with relevant references [45,48–50], the different forest zones of the Heritage Property have some similarities to the corresponding forest zones in other parts of the subtropics; the more abundant groups of soil mites have some similarities, but the Heritage Property also exhibits those characteristics.

Studies have shown that different montane forest environments have different ecological niches or ecological spaces, leading to differences in soil mite groups [10,11,43,51] and that such differences exist because soil mite trophic groups differ, and their environmental needs and adaptations lead to regional variability in group composition.

### 4.2. Variation in Vegetation along the Elevation Gradient Affects Soil Mite Community Structure

It has been shown that soil mite community structure is closely related to woodland vegetation. Litter of surface vegetation is a vital source of nutrients to woodland soil ecosystems, and its quantity and nature may have a decisive influence on soil mite community structure [52–54].

There are various vegetation types in the Heritage Property. As the altitude increases, the vegetation types also change with the change in hydrothermal conditions, and the composition and distribution of soil mites also vary more significantly. The number of individual soil mites tends to decrease and then increase with increasing altitude, and the number of groups tend to increase with increasing altitude. The variation in the number of individuals with altitudinal gradient is somewhat different from the variation in

the number of individuals of soil mites in Mount Lushan, Gaoligong Mountain, and the Santa Catalina Mountains in the USA [55–57]; the variation in the number of groups with altitudinal gradient is generally consistent with previous research [50,58,59]. This variation may be due to the number of individuals and groups of soil mites being not only influenced by altitude, but also by a combination of factors such as slope, slope orientation, and geological features [60–63]. The RBS is located in the lower reaches of the Shanmu River in the Heritage Property, which has favorable hydrothermal conditions and the highest abundance of individual soil mites. Nevertheless, located on both sides of the river, plant diversity is relatively lacking and vulnerable to disturbance by human activities; thus the diversity and richness indices are low. The EBF and DBF of the Heritage Property are typical of karst forests [64] and possess a rich number of soil groups with a high diversity index. Nevertheless, the number of rare and most rare genera groups occupies a large proportion, which is similar to the results of previous studies [8]. The predominance of rare and most rare genera in soil mites indicates a high turnover of soil mite groups, typical of unstable ecosystems [65]. The EBF and the DBF are in a more vulnerable state due to the thin soil layer in the area, which is prone to erosion, affecting the nature of soil mite communities. In these mixed coniferous and broad-leaved forests, the relatively stable dead leaves and leaves provide a practical food resource for soil mites, mitigating the impact of external disturbances on soil mites and reducing water loss [37,66]. As a result, the soil community structure is relatively stable.

Vegetation structure affects soil mite community structure by altering microclimatic conditions, the spatial distribution of soil nutrients, and creating diverse habitats [34,50]. The diversity index, richness index, and evenness index of vegetation under the vertical gradient of the heritage site all showed an increasing, then decreasing, characteristic with increasing altitude. From the sampled layers, the humus layer possessed high diversity and richness; however, the evenness index was low, similarly to the results obtained by Wang P. J. et al. [49] on the mixed evergreen deciduous broadleaf forest under the vertical gradient in the Fanjing Mountain. By comparison, foreign scholar Hasegawa, M. et al. [59] concluded that the diversity and evenness indices vary to a small extent with increasing altitude due to environmental fluctuations, but they show an overall decreasing trend. These results are inconsistent with the present study, which may be due to ecological heterogeneity disturbances, however, the seven different soil mite assemblage types in the Heritage Property, which are based on the results of cluster analysis, further suggests once again that significant microhabitat heterogeneity also exists within each vegetation zone. The EBF of the Heritage Property are typical of karst forests; they are easily disturbed and unstable. Nevertheless, it exhibits high abundance and diversity, which is quite different from the results of previous studies [8]. From this point of view, the unique ecological characteristics of karstic montane forest environments and the inconsistency of soil mite community structure characteristics with habitat gradients can also be explained.

### 4.3. Ecological Groups of Soil Mites as Indicators of the Forest Soil Environment

The ecological groups of soil mites inhabiting different forest soils occupy important environmental niches, are critical functional groups, they respond to environmental changes, and can be used as indicator organisms of the forest soil environment [14,43]. The two main groups in the soil environment are the predatory gamasid mites (Mesostigmata: Gamasina) and the oribatid mites. The type of ecological strategy of the predatory gamasid mites and the oribatid mites are often used as an essential indicator to evaluate the quality and stability of the ecological environment [35].

According to different biological characteristics, organisms have been classified into *K*-selected groups (highly competitive, high survival rate, early maturity) and *r*-selected (weakly competitive, high reproductive capacity, late maturity). In more disturbing environments, it is mainly *r*-type. The community structure of the predatory gamasid mites in the study area is mainly *r*-type [29], which differs from the stable environment where the predatory gamasid mites are predominantly *K*-type. This may be because soil mites in

Heritage property are influenced by altitude, vegetation type, and mountain tourism, so the results differ from previous studies [48,49].

Some studies have suggested that oribatid mites with low ossification, softer bodies, lower macropores, and quieter living environment requirements are suitable for living in stable environments with minor disturbances and diverse habitats, whereas oribatid mites with a higher degree of solidification of the body are suitable for living in environments with a relatively single domain and a large degree of interference [31,50]. The soil mite ecological groups in the Heritage property are mainly O and P types, indicating that the forest soil environment in the heritage site has been less disturbed overall, and that it retains a more pristine forest ecology [67].

Schneider et al. [39] classified the trophic structure of oribatid mites into four major groups, namely predators, carnivores, scavengers, omnivores; secondary decomposers; primary decomposers; and phytophagous, fungivores. Scavengers can regulate above-ground plant communities' structure, function, and succession by facilitating material cycling and energy flow in soil ecosystems [68,69]. The primary vegetation of the Heritage Property consists of zonal top-subtropical broadleaf evergreen forest, subtropical deciduous broadleaf forest, and subtropical warm coniferous broad-leaved mixed forest [4,7]. The deciduous broad-leaved mixed forest, evergreen broad-leaved forest, and coniferous broad-leaved mixed forest in the study area have a certain number of scavengers, such as the *Oppiella*, *Nothrus* and *Trichogalumna*, which to some extent reflects the interaction between soil mites and vegetation.

## 5. Conclusions

The soil mite community of the Shibing Karst World Heritage Property has a diverse number of groups, with the suborder Oribatida and genus oribatid dominating the composition, giving it a certain uniqueness. Due to the differences in vegetation types under the vertical gradient, soil mite community composition and diversity show significant variability. The similarity in soil mite community structure between vertical vegetation types is not high. Nevertheless, the analysis of dominant groups, diversity indices, and ecological groups of soil mites shows that the forest ecology of the Heritage Property is in a relatively stable state. At the same time, the dominant groups and soil mites with typical differences in biological and ecological characteristics can also be used to indicate the environmental environment of the Heritage Property. Based on the unique dolomite karst geology of the site, the vertical gradient of vegetation types, and the substantial ecological heterogeneity of the karst area, the community structure of soil mites is in the process of adjustment and dynamic change; therefore, long-term monitoring and in-depth research on the relationship between soil mite community dynamics and soil environmental factors, as well as the ecological processes involved with soil mites, are needed.

**Author Contributions:** Data curation, Y.Z.; formal analysis, Y.Z. and Q.W.; funding acquisition, H.C.; investigation, Y.F. and T.G.; software, Z.S.; writing—original draft, Y.Z.; writing—review and editing, N.X. and J.H. All authors have read and agreed to the published version of the manuscript.

**Funding:** This research was funded by National Key R&D Program of China (2016YFC0502601), Guizhou Provincial Science and Technology Foundation: (Qian Ke He Ji Chu (2020) 1Y153).

**Institutional Review Board Statement:** Not applicable.

**Informed Consent Statement:** Not applicable.

**Data Availability Statement:** The data presented in this study are available on request from the corresponding author.

**Acknowledgments:** We are grateful to the staff of the Natural Heritage Protection Administration of Shibing County for their great support to the environmental survey, and to Qiang Wei, Zheng Shi, Yifan Fei, Tong Gong, NieJia Xiao, and Ju Huang for their help in the collection and processing of experimental samples.

**Conflicts of Interest:** The authors declare no conflict of interest.

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
