# Peer review of "Characteristics of Soil Mites Communities Structure under Vegetation Vertical Gradient in the Shibing World Natural Heritage Property, China"

_forests, doi:10.3390/f13040598_

Round 1

Reviewer 1 Report

The paper's topic is relatively narrow. I really appreciate the originality of the selected region. The use of multivariate analysis would be more appropriate for statistical processing. I have a few critical comments about this article:

The article's title is too long; it is necessary to state UNESCO. It needs to be adjusted. Add UNESCO to your key words.

Introduction:

  • There is a lack of importance of mites in the eco-system, indication values of the presence of some species, relationships between species etc. - must be added
  • Move text line 45–70 from M + M
  • Define scientific hypotheses and, if necessary, set sub-objectives

Materials and Methods:

  • 1 - no source of information is given
  • Finish the nomenclature system that was employed.
  • Use SI units and replace "cm" with "m" or "mm."
  • According to which rules the sampling sites were selected (2.2.1)
  • Did you determine the proportion of organic matter in the samples?

Results:

  • It is not clear from the results what is the significance of the amount of biomass at the sampling site for the occurrence of mite taxa

Discussion:

  • Line 326–328: What is the reason for the variability in mite composition?
  • 2. Does the vegetation change with increasing altitude? Could this be the reason for the differences in mite taxa?
  • Line 340–344: Is this your finding?
  • I lack the discussion here about the relationship between mite taxa and vegetation

Conclusions:

  • Limit general phrases
  • Formulate conclusions according to hypotheses
  • Which taxa can be used as indicators of changes in the environment based on your study

Errors:

  • Line 22 - …are moderately dissimilar. The Predatory gamasid mite… Consider adding a space
  • Line 204 - "Table 4. Distributions ……" edit "Table 4. Distributions… .."

Author Response

Response to Reviewer 1 Comments

We are grateful and honored to receive your review comments. First of all, I would like to express my sincere gratitude for your hard work on the author's paper. Your constructive review comments have greatly enriched the author's research paper and provided the author with many new ideas and professional help, which is the best affirmation of the author's research work; I fully accept your review comments, according to your relevant suggestions, the author has made significant adjustments and changes in the corresponding parts of the newly submitted paper, as detailed below.

Point 1: The article's title is too long; it is necessary to state UNESCO. It needs to be adjusted. Add UNESCO to your keywords.

Response 1: Thank you for your sincere advice. The author has made the following changes for the Title, Abstract, and Keywords sections.

(1) The author has optimized and revised the paper's title to Characteristics of Soil Mites Communities Structure under vertical vegetation gradient in the Shibing World Natural Heritage Property, China.

(2) The author has written in the first part that "World Heritage is a cultural and natural heritage of outstanding and universal value, recognized by UNESCO and the World Heritage Committee as the common heritage of all humanity, and natural heritage is more treasured than cultural heritage because of its fragility and non-renewability."

The specific changes can be found in the revised manuscript in Lines 5-3.)

(3) Add UNESCO to the keywords.

Point 2: Introduction:

There is a lack of importance of mites in the eco-system; indication values of the presence of some species, relationships between species, etc. - must be added.

Move text lines 45–70 from M + M.

Define scientific hypotheses and, if necessary, set sub-objectives.

Response 2: Thank you for your sincere advice. For the Introduction section, the author has made the following changes.

The author has carefully considered your comments and entirely agrees with you. Hence the author has rewritten the introduction to this paper.

In the first paragraph, the author explains the background of the research, starting with the importance of the World Heritage Property, followed by a discussion of the significance of the Shibing World Natural Heritage Property, followed by an extensive introduction to the uniqueness of the Shibing Karst World Natural Karst Forest, highlighting the importance of forest biodiversity in the heritage property, and through a comparison of the literature, drawing out the lack of subsurface biodiversity research.

In the second paragraph, the author focuses on the relationship between soil mite communities and karst forests and presents the key scientific questions to be addressed in this paper (differences and changes in soil mite community structure under different vegetation types).

In the third paragraph, the author briefly explained the context and significance of this paper and discussed how to address the critical scientific questions raised. In the third paragraph, the author briefly describes the context and importance of the research and outlines how to manage the pressing scientific questions raised.

The author believes that the revised introduction has been improved to a certain extent compared to the previous one.

The specific changes can be found in the revised manuscript on Lines 36-89.)

Point 3: Materials and Methods:

(1) 1 - no source of information is given

(2) Finish the nomenclature system that was employed.

(3) Use SI units and replace "cm" with "m" or "mm."

(4) According to which rules the sampling sites were selected (2.2.1)

(5) Did you determine the proportion of organic matter in the samples?

Response 3: Thank you for your sincere advice. The author has made the following changes to the Materials and Methods section.

(1) In the section on the overview of the study area, the author has made changes and improvements based on your comments and has included relevant references as sources of information for the paper.

(2) The works consulted by the author in naming soil mites are Pictorial Keys to Soil Animals of China, Acarology, A manual of Acarology (3rd edition), and Soil Gamasid Mites in Northeast China. These references generally agree with the works consulted by the authors in their identification, as they also contain the corresponding nomenclature for soil mites.

The specific changes can be found in the revised manuscript on Lines 138-140.)

(3) The author has made the appropriate changes to use SI units throughout.

(4) The author has refined the sampling design by setting up six 15 cm x 15 cm sample points in an "S" or serpentine sampling pattern in a relatively stable area of about 10 m. After removing the dead leaves from the surface, the 100 mm (D) × 64 mm (H) cylindrical swivel knife was used for sampling.

(5) The proportion of organic matter in the samples could not be determined as the corresponding physicochemical soil samples were not collected in this paper.

Point 4: Results:

It is not clear from the results the significance of the amount of biomass at the sampling site for the occurrence of mite taxa.

Response 4: Thank you for your sincere advice. The author has made the following changes to the Results.

The author apologizes most sincerely for not being able to understand your question clearly. The author has tried to interpret your question as follows: Was the mass of the soil mites weighed? or whether to calculate individual densities of soil mites?

Firstly for testing the mass of the mites, as they are tiny and light, all weighing equipment is not sufficient for the author to achieve such a goal.

Secondly, this paper does not calculate the individual density of mites because all the sampling methods, including different soil layers and points, use the same cylindrical swivel knife. Suppose the density is converted according to the area. In that case, it presents the same variation pattern and number of individuals, so the author has given up calculating the individual thickness of mites. If the need for this section is confirmed, the author is fully prepared to add it later.

Point 5: Discussion:

(1) Line 326–328: What is the reason for the variability in mite composition?

(2) Does the vegetation change with increasing altitude? Could this be the reason for the differences in mite taxa?

(3) Line 340–344: Is this your finding?

(4)I lack the discussion here about the relationship between mite taxa and vegetation

Response 5: Thank you for your sincere advice. The author has made the following changes to the Discussion.

(1) Based on relevant studies and the present study results, we conclude that the changes in soil mite composition are closely related to vegetation type. In contrast, the author, in the discussion, combined with previous relevant studies, concluded that the vegetation in the Heritage Property is diverse. With the rise in altitude, the vegetation type also changes with the change in hydrothermal conditions, and the composition and distribution of soil mites also vary more significantly. The number of individuals of soil mites tended to decrease and then increase with increasing altitude, and the number of groups showed a tendency to increase with increasing altitude.

Therefore, the soil mite community structure is closely related to the vegetation of the forest floor. At the same time, the change in hydrothermal conditions with the increase in altitude also impacts the soil mite community structure.

The specific changes can be found in the revised manuscript on Lines 413-464.)

  • As altitude increases and hydrothermal conditions change, the type of mountain vegetation changes accordingly. In this research, the author discusses the trophic groups of soil mites that play an essential role in the soil food web. Schneider (2004) et al. classified the trophic structure of oribatid mites into four major groups: predators, carnivores, scavengers, omnivores; secondary decomposers; primary decomposers and phytophagous, fungivores. Scavengers can regulate above-ground plant communities' structure, function, and succession by facilitating material cycling and energy flow in soil ecosystems. The primary vegetation of the Heritage Property consists of zonal top-subtropical broadleaf evergreen forest, subtropical deciduous broadleaf forest, subtropical warm coniferous broad-leaved mixed forest, and the deciduous broad-leaved mixed forest, evergreen broad-leaved forest and coniferous broad-leaved mixed forest in the study area have a certain number of scavengers, such as the OppiellaNothrus and Trichogalumna, which to some extent reflects the interaction between soil mites and vegetation. The conclusions drawn in this section have been inserted in the revised manuscript with relevant references.

The conclusions drawn in this section have been inserted in the revised manuscript with relevant references.

The specific changes can be found in the revised manuscript on Lines 413-464.)

(3) The author must sincerely apologize that the author's paper is not designed to address the role of the slope, slope orientation, and geological features on mites. The author's speculative conclusion is based on previous research, which may not be reliable. In later studies, the authors will investigate the effects of slope, slope orientation, and geological features on soil mites.

The specific changes can be found in the revised manuscript on Lines 427-430.)

  • The authors revised and refined the discussion of the relationship between soil mites and vegetation in the revised manuscript.

The specific changes can be found in the revised manuscript on Lines 413-464.)

Point 6: Conclusions:

(1) Limit general phrases.

(2) Formulate conclusions according to hypotheses.

(3) Which taxa can be used as indicators of changes in the environment based on your study.

Response 6: Thank you for your sincere advice.The author has made the following changes to the Conclusions.

  • The author has corrected all the formatting errors that you have raised and has also made some minor corrections to the English language due to the magnitude of this revision and adjustment, which may cause you a lot of inconvenience in reviewing the manuscript, for which the author apologizes most sincerely.
  • Based on the author's research and previous studies, we can use the dominant groups of soil mites as indicators of environmental change. In this study, Perscheloribatesand Scheloribates are the dominant groups in the study area. We can use these two soil mite species as essential indicators to detect environmental changes in forest soils.

The specific changes can be found in the revised manuscript at:Lines 502-518.)

Point 7: Errors:

(1) Line 22 - …are moderately dissimilar. The Predatory gamasid mite… Consider adding a space

(2) Line 204 - "Table 4. Distributions ……" edit "Table 4. Distributions… .."

Response 7: Thank you for your sincere advice.

The author has corrected all the formatting errors you have raised in the text. Following your suggestions, this paper has been substantially revised, and the English writing has been improved and refined. The author apologizes most sincerely for any inconvenience this may cause you in reviewing your paper.

At this point, the author must once again thank you for your valuable corrections! I hope you will let me know if you find any shortcomings again during the review process, and I will take them seriously. Thank you again!

Reviewer 2 Report

Reviewer

Manuscript intitled: “Characteristics of Soil Mite Community Structures under Different

Vegetation Vertical Zones in Shibing Karst World Natural Heritage Site, China”

Authors: Yuanyuan Zhou, Qiang Wei, Niejia Xiao, Ju Huang, Tong Gong, Yifan Fei, Shi Zheng, Hu Chen

The authors present to us the diversity and differences of soil mite community structures under different vegetation vertical zones in Shibing Karst World Natural Heritage Site. The four typical vegetation zones were investigated, from acarological point of view: coniferous broad-leaved mixed forest, evergreen broad-leaved forests, deciduous broad-leaved forests, and river beach scrub. A total of 10,563 soil mites were collected in this region, belonging to 3 orders, 67 families, 137 genera. Perscheloribates and Scheloribates are the dominant mite genera in the region. The results of cluster analysis show that the soil mite communities are different in different vegetation zones, all of which are moderately dissimilar. The study shows that the soil mites’ community structure and composition in the Heritage Site presents a gradient difference with the vegetation band spectrum. Based on the unique habitat of the subtropical canyon karst of the heritage site, the community structure of soil mites will be in the process of adaptative adjustment, hence long-term dynamic monitoring and in-depth study are required on soil mite community structure in the heritage site.

The information presented in the manuscript is original, well-structured and documented. It is obvious that the authors collected a lot of mite samples (10563 individuals). Any information regarding the soil mite fauna from unstudied ecosystems are precious, but I strongly recommend the identification of the mite material till species level. Most often, in these unique habitats it is possible to identify new species for science or other bioindicator species for investigated ecosystems. That’s why I consider this manuscript as a generalist one.

  • The title of the manuscript reflects the content, but I strongly recommend a little change.
  • The manuscript is sustained by a suitable literature, but I recommend adding some references!
  • The methodology chapter is detailed described. The authors used statistical analysis in a proper manner!
  • The results and discussions are proper described and presented.
  • The tables and figures reveal properly the described results/data.
  • References:

Please, check if the all references were found in the manuscript and vice versa.

Please, follow the instruction for the authors for references.

On the other hand major comments must be added:

  1. The title: I recommend to modify the title, as following: Characteristics of Soil Mite Communities Structures under vegetation vertical gradient ...
  2. Lines 14-15: Please, rephrase! The phrase has no sense. See the comment from the title!
  3. Line 31- Keywords: Why you used capital letters?
  4. Line 34- Introduction: the informations regarding the importance of soil mite fauna in different types of ecosystems are too poor. The authors does not mentions if the mite fauna was studied in this region. The objectives/hypotheses of the studied are not clearly presented. I recommend reconsidering the introduction, highlighting the relationship between soil mite fauna and the vertical distribution of forests ecosystems.
  5. Lines 77-79, 85: please insert references!
  6. Line 92: I recommend inserting a map and some photos with investigated forest ecosystems.
  7. Line 103: Why you collected separately the leaf litter? I don't understand. Which is the number of samples taken with cylindrical swivel knife? You used to methods of mite collection, but in the results you treat both samples as equal. In order to surprise the vertical distribution of soil mites, you must to collect whole sample: litter, fermentation,humus and soil layers! Because your study is focused on influence of vertical distribution of vegetation of soil mites, it is proper to provide the vegetation coverage of each sample!
  8. Lines 118-122: Insert the references! The dominance classes for the identified soil mites were: eudominant species with dominance over 10% (D5); dominant species with dominance between 5.1-10% (D4); subdominant species with dominance between 2.1 - 5% (D3); recedent species with dominance between 1.1 - 2% (D2) and subrecedent species with dominance under 1.1% (D1) (Engelmann, 1978).
  9. Line 138: Please, be careful at text format! Insert breaks between the words or delete the inutile spaces!
  10. Line 164: This software is not inserting at references!
  11. Lines 172-178: I strongly recommend another formulation of results. I consider that these results must be presented taking into consideration the three orders Mesostigmata, Trombidiformes and Sarcoptiformes!
  12. Lines 187-192: Taking into consideration the table 4, in this passage I consider that is necessary to discuss how significant were these differences (p value?)
  13. Line 203, Table 3: Please check carefully the scientific names of families and genera!
  14. Line 206: How you define the horizontal and vertical distribution? If your manuscript is focused on vertical gradient of vegetation, here you discussed about the distribution on soil habitat! How it is possible to discussed about the vertical distribution of mites in soil, when you used two types of collecting methods: cylindric swivel knife and 15X15 cm sampling area for litter? Why you collect separately the litter?
  15. Lines 211-212: On the material and methods chapter, please define what the meaning of Layer A is and B.
  16. Line 296: But in the table 8, you don't list 86 genera!
  17. Lines 300-301: But dominant genera are grouped in which category? And why?
  18. Line 329: These differences between investigated forest ecosystems are due to the different soil properties, to the different environmental conditions, to the bio-chemical structure of the litter-fermentation layer, of the different level of soil decomposition. All these informations must be argued with literature!
  19. Line 336: From geographical point of view and climatic conditions this area could be compared with a region from China? It is proper this comparison?
  20. Lines 341-343: This is an assumption! It is not a certain results obtained by the authors!
  21. Lines 375-376: Please insert some references!
  22. Line 421: The correct name is: Manu, M.

I strongly recommend The English spelling and checking by a native speaker!

The text format must be checked! There are many spelling mistakes!

All these comments are inserting in the manuscript!

Author Response

Response to Reviewer 2 Comments

We are grateful and honored to receive your review comments. First of all, I would like to express my sincere gratitude for your hard work on the author's paper. Your constructive review comments have greatly enriched the author's research paper and provided the author with many new ideas and professional help, which is the best affirmation of the author's research work; I fully accept your review comments, according to your relevant suggestions, the author has made significant adjustments and changes in the corresponding parts of the newly submitted paper, as detailed below.

Point 1: The title: I recommend modifying the title, as follows: Characteristics of Soil Mite Communities Structures under vegetation vertical gradient ...

Response 1: We thank you for your sincere advice.

We have revised the title according to your suggestion to Characteristics of Soil Mites Communities Structure under vegetation vertical gradient in the Shibing World Natural Heritage Property, China.

Point 2: Lines 14-15: Please, rephrase! The phrase has no sense. See the comment from the title!

Response 2: Thank you for your sincere advice, and we have made the following changes to the abstract section.

The author has rewritten the background and purpose of the research in the abstract section; in addition, the author has made significant changes and adjustments to the findings in the abstract area.

Point 3: Line 31- Keywords: Why you use capital letters?

Response 3: Thank you for your sincere advice.

The author has double-checked the keywords in this paper and confirmed through literature comparison that "Shibing Karst World Natural Heritage Property" is a proprietary name and is therefore capitalized.

Point 4: Line 34- Introduction: the information regarding the importance of soil mite fauna in different types of ecosystems is too poor. The authors do not mention if the mite fauna were studied in this region. The objectives/hypotheses of the study are not presented. I recommend reconsidering the introduction, highlighting the relationship between soil mite fauna and the vertical distribution of forests ecosystems.

Response 4: Thank you for your sincere advice. The author has made the following changes to the Introduction.

The author has carefully considered your comments and entirely agrees with you, so the author has rewritten the introduction to this paper. In the first paragraph, the author explains the background of the research, starting with the importance of the World Heritage Property, followed by a discussion of the significance of the Shibing Karst World Natural Heritage Property, followed by an extensive introduction to the uniqueness of the Shibing Karst forest, highlighting the importance of the forest biodiversity of the Heritage Property, and through a comparison of the literature, the volume of subsurface biodiversity studies. The significance of the study of soil mites is finally discussed.

In the second paragraph, the author focuses on the relationship between soil mite communities and karst forests and presents the critical scientific question to be addressed in this paper (differences and changes in mite community structure under different vegetation types).

In the third paragraph, the author briefly explains the context and significance of the study and discusses how to address the critical scientific questions raised. The author believes that the revised introduction has been improved to some extent compared to the previous one.

(Specific changes can be found in the revised manuscript in Lines 36-39)

Point 5: Lines 77-79, 85: please insert references!

Response 5: Thank you for your sincere advice.

Relevant references have been added to the revised paper.

Point 6: Line 92: I recommend inserting a map and some photos with investigated forest ecosystems.

Response 6: Thank you for your sincere advice.

The author agrees with your comments and has considered whether to include GIS maps of the sampling sites or images of the sample sites when writing this paper and has confirmed that I have added the latitude and longitude coordinates of the sampling sites to the Heritage Property vector map. This verticality of mountain elevation makes the author's sampling points very dense in the Heritage Property. There are difficulties obtaining the relevant maps, so the author gave up making the relevant maps and presented the sampling point information in a table instead.

As for the photos of the distribution of the relevant vegetation types, since the study sample sites are World Natural Heritage Property, which is highly authoritative and has more relevant studies, the author does not consider it necessary to add the photos of vegetation types repeatedly in this paper. So the author has not added photos of the relevant forest ecosystems. I hope that the reviewer will support and understand the author's idea.

Point 7: Line 103: Why do you collect separately the leaf litter? I don't understand. Which is the number of samples taken with a cylindrical swivel knife? You used two methods of mite collection, but you treat both samples as equal in the results. To surprise the vertical distribution of soil mites, you must collect the whole sample: litter, fermentation,humus and soil layers! Because your study is focused on influence of vertical distribution of vegetation of soil mites, it is proper to provide the vegetation coverage of each sample!

Response 7: Thank you for your sincere advice.

The author describes the experimental design in detail here. As to why the litter layer was collected, the author may have inaccurately stated that it should be the humic layer, as the heritage forest has a very thick humic layer. The author picked up the fresh fallen leaves from the surface layer, scraped off some of the dead branches and leaves within the 15*15 cm sample point when sampling, and then collected three layers with A 100 mm (D) × 64 mm (H) cylindrical swivel knife. The first layer was the humic layer. The author confirmed that all samples were collected with the same cylindrical swivel knife. The humic layer, or 3 layers of soil, was composed to account for differences in soil mites of different vegetation types under the same soil layer.

Point 8: Lines 118-122: Insert the references! The dominance classes for the identified soil mites were: dominant species with dominance over 10% (D5); dominant species with dominance between 5.1-10% (D4); subdominant species with dominance between 2.1 - 5% (D3); recedent species with dominance between 1.1 - 2% (D2) and subrecedent species with dominance under 1.1% (D1) (Engelmann, 1978).

Response 8: Thank you for your sincere advice.

For selecting the threshold for classifying dominant groups, the author has downloaded the literature you recommended and believes that it is also very feasible. Still, the author prefers to stick with this classification because in reading the extensive literature on soil fauna, the author found that the majority of researchers used the same classification criteria as the author. The author believes that this will help to provide a comparison for future studies, while the same classification criteria will also facilitate the author's comparisons. However, the author has to admit that there was an error in presenting the author's classification criteria between rare and most rare groups, which the author has corrected.

Point 9: Line 138: Please, be careful at text format! Insert breaks between the words or delete the inutile spaces!

Response 9: Thank you for your sincere advice.

The author has revised the notes of the formulas in this paper and adjusted the formula numbers (1)(2)... to (I), (II)..., to avoid duplication with the subheadings.

(Specific changes can be found in the revised manuscript on Lines 151-173)

Point 10: Line 164: This software is not inserting references!

Response 10: Thank you for your sincere advice.

The author has added references to software and statistical tests.

Point 11: Lines 172-178: I strongly recommend another formulation of results. I consider that these results must be presented considering the three orders Mesostigmata, Trombidiformes, and Sarcoptiformes!

Response 11: Thank you for your sincere advice. Thank you for your sincere advice. The author has made the following changes in the section on results section.

The author agrees with your suggestion that, following the critical scientific question of the paper, the author should present the overall picture of this investigation. Therefore, the author has added a section on the composition and dominance of soil mites, which is shown on the Order and Suborder level. In addition, two new figures have been added to present the variation in the number of soil mites genera and individuals and the variation in the dominance of groups under different vegetation patterns.

(Specific changes can be found in the revised manuscript on Lines 197-224)

Point 12: Lines 187-192: Taking into consideration table 4 in this passage I consider that is necessary to discuss how significant were these differences (p value?)

Response 12: Thank you for your sincere advice.

The author describes and discusses the statistical tests for the number of soil mite genera and individuals under different vegetation types.

(Specific changes can be found in the revised manuscript on: Lines 237-243 and Lines 268-278)

Point 13: Line 203, Table 3: Please check the scientific names of families and genera carefully!

Response 13: Thank you for your sincere advice.

The author has double-checked the Latin spelling of all soil mite species.

Point 14: Line 206: How you define the horizontal and vertical distribution? If your manuscript is focused on vertical gradient of vegetation, here you discussed about the distribution on soil habitat! How it is possible to discussed about the vertical distribution of mites in soil, when you used two types of collecting methods: cylindric swivel knife and 15X15 cm sampling area for litter? Why you collect separately the litter?

Response 14: Thank you for your sincere advice.

The author sincerely apologizes to you for the lack of clarity in the presentation. The author addresses three questions in this paper. Firstly, what is the soil mite community structure in the karst forest ecosystem of the Heritage Property? Secondly, what is the impact of different vegetation types on soil mite community structure? Third, does this effect vary, and what are the specific patterns of variation? The author sets a vertical gradient that is the gradient of different vegetation types. For the differences between the different vegetation types, the author shows them in three separate soil layers. The author may have been unclear about these three soil layers in the previous manuscript. Still, I confirm that they are soil mite samples from the humic layer, soil upper layer, and soil lower layer. I use all three concepts of humic layer, soil upper layer, and soil lower layer in the newly revised manuscript without abbreviations or acronyms. The author also confirms that all samples were taken with a cylindrical swivel knife of the same size as the soil.

(Specific changes can be found in the revised manuscript at:Lines 129-130 and Lines299-236)

Point 15: Lines 211-212: On the material and methods chapter, please define what the meaning of Layer A is and B.

Response 15: Thank you for your sincere advice.

The author uses all three concepts of humic layer, soil upper layer, and soil lower layer in the newly revised manuscript without abbreviations or acronyms. 

Point 16: Line 296: But in the table 8, you don't list 86 genera!

Response 16: Thank you for your sincere advice.

For the 86 genera not listed, the author has updated Table 3 with the addition of orders and suborders, and the 86 genera of the oribatid mites we can view in the updated Table

Point 17: Lines 300-301: But dominant genera are grouped in which category? And why?

Response 17: Thank you for your sincere advice.

Concerning the trophic structure of oribatid mites, the author's current survey identified 86 genera of mites in the suborder Oribatida of the order Sarcoptiformes. Still, based on the literature of relevant researchers, only those genera presented in the table have information confirming that they belong to the corresponding trophic groups, some of which occupy multiple trophic levels. At the same time, the dominant group of soil mites described above, although significant in terms of the number of individuals, does not correspond precisely to the trophic group, i.e., the genera that are the dominant group have no information on the trophic level occupied in the food web.  

Point 18: Line 329: These differences between investigated forest ecosystems are due to the different soil properties, to the different environmental conditions, to the bio-chemical structure of the litter-fermentation layer, of the different level of soil decomposition. All these informations must be argued with literature!

Response 18: Thank you for your sincere advice.

The author has changed the language description and added relevant references.

(Specific changes can be found in the revised manuscript at:Lines 383-412 and Lines 413-464)

Point 19: Line 336: From geographical point of view and climatic conditions this area could be compared with a region from China? It is proper this comparison?

Response 19: Thank you for your sincere advice.

After careful consideration, the author agrees with the reviewer that with fewer study sites and too large scale comparisons, the conclusions obtained are often less scientific and reliable, so the author has now replaced the relevant comparative support materials, selecting one of the World Natural Heritage Property in the Southern China Karst Series - Fanjing Mountain for the relevant study, choosing the same non-karst forest in Guizhou Province, Chishui Alsophila Natural Reserve, for the comparison, and selecting mountain forests in the same subtropical climate zone - Wuyi Mountain, Huangshan Mountain, and Mitra National Park for the comparison of forest soil mites.

Point 20: Lines 341-343: This is an assumption! It is not a certain results obtained by the authors!

Response 20: Thank you for your sincere advice.

The author has combined relevant references and found relevant research from previous studies. In combination with previous research, I have concluded that the reason for the differences in community structure between the Shibing Karst World Natural Heritage Property and other study areas is due to differences in the trophic groups of soil mites, whose needs and adaptations to the environment lead to regional differences in groups composition. Therefore, the author considers this to be a result of this study and has changed the presentation and inserted relevant references to make the findings more convincing.

Specific changes can be found in the revised manuscript atLines 383-412

Point 21: Lines 375-376: Please insert some references!

Response 21: Thank you for your sincere advice.

References have been inserted in the appropriate places in the paper

Point 22: Line 421: The correct name is: Manu, M.

Response 22: Thank you for your sincere advice.

The author has corrected all the formatting errors you have raised in the text.Following your suggestions, this paper has been substantially revised, and the English writing has been improved and refined. The author apologizes most sincerely for any inconvenience this may cause you in reviewing your paper.

At this point, the author must once again thank you for your valuable corrections! I hope you will let me know if you find any shortcomings again during the review process, and I will take them seriously. Thank you again!

Round 2

Reviewer 1 Report

The authors made corrections to the manuscript and significant development in the presented results, in accordance with the recommendations of the reviews. The article has a much higher substantive value, the resources of the presented results and their quality. In this regard, minor corrections may still be introduced, however, I recommend the article to be accepted for publication, and the suggested corrections may be introduced at the editorial preparation stage. Good luck! 

Author Response

Thank you very much for your help during this time and all your efforts in this paper. Your constructive comments have been invaluable to me, and I can see from your comments that your rigorous and conscientious approach to research and your professional and extensive knowledge base, which will be the best gift to me on my research journey. And I would like to thank you most sincerely.

At this point, the author must once again thank you for your valuable corrections! I hope you will let me know if you find any shortcomings again during the review process, and I will take them seriously. Thank you again!

Reviewer 2 Report

The minor comments or observations were included in revised manuscript!

Please, pay attention to the scientific names of genera and fammilies! There are few mistakes in table 3. Check using Fauna europaea or the identification keys!

Author Response

Response to Reviewer 2 Comments

Thank you very much for your help during this time and all your efforts in this paper. Your constructive comments have been invaluable to me, and I can see from your comments that your rigorous and conscientious approach to research and your professional and extensive knowledge base, which will be the best gift to me on my research journey. And I would like to thank you most sincerely.

I fully accept your review comments. According to your relevant suggestions, the author has made significant adjustments and changes in the corresponding parts of the newly submitted paper, as detailed below.

Point 1: Question about the initials of The Predatory gamasid mite structure in line 37.

Response 1: Thank you for your sincere advice.

The author has Predatory amended to predatory.

Point 2: About Main plants in Table 1.

Response 2: Thank you for your sincere advice.

The "s" have been deleted and revised to read: Tree layer, Shrub layer, and Herb layer.

Point 3: Please, pay attention to the scientific names of genera and families! There are few mistakes in table 3. Check using Fauna europaea or the identification keys!

Response 3: Thank you for your sincere advice.

The Rliodacaridae has been revised to Rhodacaridae.

The Ologainasidae has been revised to Ologamasidae.

Revise the groups in Table 3 to taxa.

Point 4: Revised following the contents of the revised manuscript.

Response 4: Thank you for your sincere advice.

The author has revised the paper in terms of formatting, language, and content in light of the revised manuscript's content and the reviewer's suggestions.

At this point, the author must once again thank you for your valuable corrections! I hope you will let me know if you find any shortcomings again during the review process, and I will take them seriously. Thank you again!
